# Bone mineral density and related clinical and laboratory factors in peritoneal dialysis patients: Implications for bone health management

**Rami Tamimi**[1]☯, **Amjad Bdair**[1]☯, **Ahmad Shratih**[1]☯, **Mazen Abdalla**[1,2], **Alaa Sarsour**[3], **Zakaria Hamdan**[1,4]*, **Zaher Nazzal**[1]*

**1** Department of Medicine, Faculty of Medicine and Health Sciences, An-Najah National University. Nablus, Palestine, **2** Department of Orthopedics, An-Najah National University Hospital, Nablus, Palestine, **3** Kidney and Dialysis Section, An-Najah National University Hospital, Nablus, Palestine, **4** Department of Internal Medicine, An-Najah National University Hospital, Nablus, Palestine

☯ These authors contributed equally to this work.
* znazzal@najah.edu (ZN); z.hamdan@najah.edu (ZH)

**Data Availability Statement:** All relevant data are within the manuscript and its Supporting Information files.

## Abstract

### Background

End-stage renal disease (ESRD) patients often experience accelerated bone turnover, leading to osteoporosis and osteopenia. This study aimed to determine the prevalence of osteoporosis in Peritoneal Dialysis (PD) patients using bone mineral density (BMD) measurements obtained through dual-energy X-ray absorptiometry (DEXA) scan and to explore any possible associations with clinical and biochemical factors.

### Methods

In this cross-sectional study, we enrolled 76 peritoneal dialysis patients from the dialysis center at An-Najah National University Hospital in Nablus, Palestine. We used the DEXA scan to measure BMD at the lumbar spine and hip, with values expressed as T-scores. We conducted a multivariate analysis to explore the relationship between BMD and clinical and biochemical parameters.

### Results

Over half (52.6%) of the PD patients had osteoporosis, with a higher prevalence observed among patients with lower BMI (p<0.001). Higher alkaline phosphatase levels were found among osteoporotic patients compared to non-osteoporotic patients (p = 0.045). Vitamin D deficiency was also prevalent in this population, affecting 86.6% of patients. No significant correlation was found between 25 vitamin D levels and BMD. No significant correlation was found between Parathyroid hormone (PTH) levels and BMD.

**Funding:** The authors received no specific funding for this work.

**Competing interests:** The authors have declared that no competing interests exist.

## Conclusion

A notable proportion of PD patients experience reduced BMD. Our study found no correlation between vitamin D levels and BMD, but it highlighted the significant vitamin D deficiency in this population. Furthermore, our analysis indicated a positive correlation between BMI and BMD, especially in the femoral neck area. This underscores the significance of addressing bone health in PD patients to mitigate the risk of fractures and improve their overall well-being.

## Introduction

Chronic kidney disease (CKD) is a significant public health concern and a major cause of premature death. End-Stage–Renal Disease (ESRD) is the result of CKD and is defined as GFR less than 15 ml/min, which ultimately needs kidney replacement therapy (KRF), dialysis, or transplantation [1]. In 2016, Peritoneal Dialysis (PD), a type of kidney replacement therapy [2], was established in Palestine despite many challenges and limitations. It has been successfully developed over the years to meet the demand of the growing number of ESRD patients in need of kidney replacement therapy [3].

CKD patients suffered from accelerated bone turnover and decreased bone density, for which a broad range of pathophysiological states could be responsible, such as secondary hyperparathyroidism [4], hyperphosphatemia [5], decreased vitamin D synthesis [6], hypocalcemia [5], and advancing age. This makes ESRD patients more vulnerable to bone fractures than the general population [7, 8].

Bone mineral density (BMD) serves as an indicator of bone mass and mineralization [9]. According to the International Society for Clinical Densitometry (ISCD), BMD can be assessed by Dual-Energy X-ray Absorptiometry (DEXA) scan at both the postero-anterior lumber spine (L1-L4) and the hip (femoral neck or total proximal femur) [10]. The DEXA scan results can be expressed as BMD (g/cm2), Z-score, or T-score, with the T-score representing the number of standard deviations (SD) from the mean of a healthy young adult.

The 2017 KDIGO Chronic Kidney Disease-Mineral and Bone Disorder (CKD-MBD) guideline update recommends BMD testing for patients with CKD who show evidence of CKD-MBD and/or have risk factors for osteoporosis [11]. A meta-analysis of studies in CKD patients revealed a significant relationship between a low BMD and the risk of bone fracture [12], confirming the importance of BMD measurement and fracture risk prevention. Identifying low BMD in CKD patients early is crucial, particularly in candidates for kidney transplants, as treatment becomes more challenging post-transplantation [13].

Vitamin D facilitates calcium absorption in the gastrointestinal tract, ensuring sufficient calcium and phosphate levels in the bloodstream. This, in turn, supports the process of normal bone mineralization [14]. Vitamin D deficiency, characterized by a 25-hydroxyvitamin D level below 20 ng/mL, is higher among individuals with CKD than the general population[15]. Multiple factors contribute to vitamin D deficiency, such as progressive hyperparathyroidism [16], hyperphosphatemia, reduced sunlight exposure, and limited dietary intake. Vitamin D deficiency is clinically associated with various medical conditions, including increased cardiovascular morbidity [17], an elevated risk of peritonitis [18], impaired physical performance [19], and decreased bone mineral density [20].

A previous study conducted at An-Najah National University Hospital (NNUH), which included a majority of hemodialysis (HD) patients, found that 42.8% of ESRD patients had

osteoporosis and 40.2% had osteopenia [21]. This study reported no significant difference between patients on hemodialysis or peritoneal dialysis, but it acknowledged that the small sample size of PD patients may limit this result. One regional multicenter study that included 292 PD patients found that 19% of patients had osteoporosis and 36% had osteopenia [22], somewhat similar to what other studies have demonstrated [23–26]. In addition, these studies found an increased risk of reduced BMD with lower Body Mass Index (BMI). However, conflicting results were reported regarding the relation between DEXA scan measurements and biochemical markers. In this study, we aim to assess the prevalence of osteoporosis/osteopenia and evaluate the clinical and biochemical variables associated with BMD in PD patients.

## Materials and methods

### Design and population

This cross-sectional study was conducted between August 2022 and November 2022 at the dialysis center of NNUH, Palestine, the largest dialysis center offering the option of PD in the region.

The study population consisted of ESRD patients undergoing PD and were regularly followed up by NNUH physicians. A total of 76 patients undergoing PD were recruited, encompassing a significant proportion of the NNUH dialysis center's patient population. Patients on continuous ambulatory peritoneal dialysis above 18 years of age were included. Conversely, patients with a history of parathyroidectomy or malignancy, as well as patients currently taking bisphosphonates or estrogen, were excluded from our study. None of our patients were on any calcimimetic treatment or inactive vitamin D supplementation.

### Study measures

After obtaining informed consent from the participants, we extracted baseline demographic and clinical characteristics from patients and their medical records. These variables include age, gender, BMI, duration of PD, smoking status, presence of comorbidities, history of fractures, kidney transplant, history of HD, menopause status, and current medication history (alpha D3 supplements, calcium supplements, phosphate binder use, and steroids use). Laboratory and biochemical markers, including serum level of 25-hydroxyvitamin D, albumin, calcium, phosphate, alkaline phosphatase, parathyroid hormone (PTH), hemoglobin, and ferritin, were obtained via venous blood sampling at the time of their monthly follow-up visits. The blood samples were immediately sent to the laboratory for analysis upon collection and were subsequently tested on the collection day.

DEXA scan is considered one of the best available ways to assess BMD because it is noninvasive and accurate, with a short scanning time and low radiation exposure. The World Health Organization (WHO) defines osteoporosis as a T score on BMD measurement using a DEXA scan at the femoral neck -2.5 or lower, where a T score between -1.0 and -2.5 is defined as Osteopenia [27]. For this study, patients were classified as osteoporosis and non-osteoporosis groups. The osteoporosis group was defined as any patient with a T score of -2.5 or lower in at least one site, while the non-osteoporosis group was defined as any patient with a T score of above -2.5. The BMD assessment was conducted through a Dual-Energy X-ray Absorptiometry (DEXA) scan using the Hologic equipment model Discovery WI with serial number 82189. The imaging process was carried out and evaluated by experienced technicians at Al-Rahma Medical Center, Nablus, within the same month as the blood tests.

The Fracture Risk Assessment Tool (FRAX), developed by the World Health Organization Collaborating Centre in Sheffield, UK, is an online web-based algorithm that estimates the individualized 10-year probability of hip and major osteoporotic fracture (MOF; hip, clinical

spine, distal forearm, and proximal humerus). The algorithm incorporates seven dichotomous clinical risk factors: age, sex, BMI, and, optionally, a BMD measurement [28]. FRAX has been validated for appropriate use in Palestinian patients at high risk of developing osteoporosis.

The Elecsys kit was used to determine the level of 25 vitamin D in the body. Vitamin D deficiency was defined as a 25-hydroxyvitamin D level below 20 ng/mL [29]. The Elecsys kit also measured the Parathyroid hormone (PTH) level. Around 5-10cc of venous blood was drawn by trained nurses at the NNUH dialysis center during patients' monthly follow-up visits, and the samples were sent to the laboratory for analysis.

## Analysis plan

We processed the collected data using the Statistical Package for the Social Sciences version 26 (IBM Corp., Armonk, NY, United States). Subsequently, we described the findings as frequency tables and suitable charts for categorical variables, while continuous variables were presented using mean and standard deviation (SD) or median and range values. We used the Shapiro-Wilk test to assess for data normality. For analytical analysis, patients were categorized into osteoporosis and non-osteoporosis groups. Univariate statistical analysis was performed, utilizing relevant significance tests such as Mann-Whitney U, Chi-Square, and independent 2-sample tests. Subsequently, a multivariate analysis was conducted using binary logistic regression, incorporating variables that demonstrated significance in the univariate analysis and additional variables recommended by existing literature. The variables included in the regression model were age, gender, BMI, duration of dialysis, calcitriol supplementation, calcium supplementation, sevelamer supplementation, 25 Vitamin D, alkaline phosphatase, parathyroid hormone, albumin, calcium measured, and phosphate. Multiple linear regression and Spearman's correlation coefficients were used to investigate correlations. The variables deemed significant in the univariate analysis were included in the multiple linear regression (BMI, Sevelamer Dose, Albumin, Phosphate). Statistical significance was set at p-value ≤0.05.

## Ethical consideration

The study and its associated experimental protocols, such as performing DEXA scans and drawing blood, were approved by the Institutional Review Board committee of An-Najah National University [Reference #: Med. Feb. 2022/21]. Appropriate permissions were obtained from the hospital and Rahma Medical Center. The study adhered to the principles outlined in the Declaration of Helsinki and complied with applicable national guidelines and regulations. Patients were given the option to voluntarily participate in the study, with a comprehensive explanation of its purpose, objectives, and potential risks. To ensure confidentiality, no personal information was collected, and codes rather than names identified patients. Access to the collected data was restricted to the research team for research purposes only. All patients provided written informed consent before participating.

## Results

### Background characteristics of the patients

A total of 76 Peritoneal Dialysis patients were recruited to participate in this study. The mean age of our participants was 49.5 years (SD = 16.5), with 68% (n = 52) under the age of 60. Approximately 54% (n = 41) of the participants were males, 46% (n = 35) had normal BMI, and 83% had been on PD for over a year. Among the participants, 75% (n = 57) had hypertension and 33% (n = 25) had Diabetes mellitus. Around 8% (n = 6) of our patients reported a

**Table 1. Clinical and laboratory characteristics of the participants.**

| | Frequency (%) | Mean ± SD |
|---|---|---|
| **Age** | | 49.5 ± 16.5 |
| Age ≥ 60 years | 24 (31.6%) | |
| Age <60 years | 52 (68.4%) | |
| **Gender** | | |
| Male | 41 (53.9%) | |
| Female | 35 (46.1%) | |
| **Body Mass Index** | | 25.4 ± 5.5 |
| Underweight | 6 (7.9%) | |
| Normal | 35 (46.1%) | |
| Overweight | 35 (46.05) | |
| **Duration of Peritoneal Dialysis** (*Months*) | | 23.0 ± 15.0 |
| **Hypertension** (*Yes*) | 57 (75.0%) | |
| **Diabetes Mellitus** (*Yes*) | 25 (32.9%) | |
| **Smoking History** (*Yes*) | 27 (35.5%) | |
| **History of Hemodialysis** (*Yes*) | 57 (75.0%) | |
| **History of Kidney Transplant** (*Yes*) | 13 (17.1%) | |
| **History of Fractures** (*Yes*) | 6 (7.9%) | |
| **Menopause Status** (*Yes*) | 18 (51.4%) | |
| **Steroid Use** (*Yes*) | 4 (5.3%) | |
| **1,25 Vit D3 Supplementation (calcitriol)** (*mcg/day*) | | 0.5 ± 0.4 |
| **Calcium Supplementation** (*mg/day*) | | 1869.8 ± 1112.6 |
| **Sevelamer Binder Supplementation** (*mg/day*) | | 3282.8 ± 1118.1 |
| **Alkaline Phosphatase** (*U/L*) | | 122.3 ± 67.1 |
| **Parathyroid Hormone** (*pg/ml*) | | 439.4 ± 323.2 |
| **Albumin** (*g/dl*) | | 3.65 ± 0.43 |
| **Calcium Measured** (*mg/dl*) | | 9.1 ± 0.71 |
| **Phosphate** (*mg/dl*) | | 5.46 ± 1.56 |
| **Ferritin** (*ng/ml*) | | 576.7 ± 588.8 |

history of fractures while on dialysis. Table 1 summarizes the patients' baseline demographic, biochemical, and clinical features.

Vitamin D lab results demonstrated that around 86.8% (n = 66) [95%CI: 77.1% - 93.5%] of our patients suffer from 25 vitamin D deficiency, with a median of 8.7 *(ng/ml)*.

## Bone mineral density

The overall prevalence of osteoporosis among the participants was 52.6% [n = 40, 95% CI: 40.8%-64.2%]; according to the site, 41% of the patients had osteoporosis in the femoral neck (Median T-score = -2.3) and 38% had osteoporosis in the lumbar spine (Median T-score = -2.1) (Fig 1). According to the FRAX tool, the median 10-year probability of developing a major osteoporotic fracture was 4.1%, with scores ranging up to 52%. In comparison, the median FRAX score for risk of hip fractures was 1.5% (0.1%-49.0%).

A univariate analysis was carried out to examine the relationship between patients' BMD status and their demographic and clinical attributes. BMI was significantly higher in non-osteoporotic patients than in osteoporotic patients (p = 0.001); this relationship remained significant after multivariate analysis, as demonstrated in Table 2.

Multivariate analysis revealed slightly higher alkaline phosphatase levels among osteoporotic patients than non-osteoporotic patients (p = 0.045). Although the age of osteoporotic

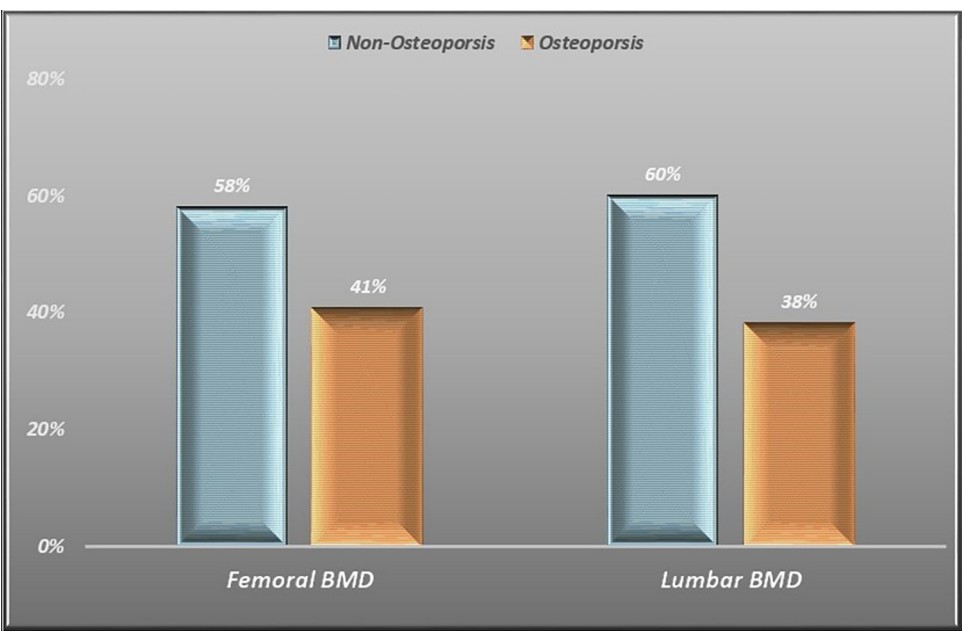

**Fig 1. Bone mineral density distribution.**

patients tended to be less than that of non-osteoporotic patients, this difference was not statistically significant even after adjusting for confounders (p = 0.108). Regarding gender, males exhibited higher levels of osteoporosis than females, but this relationship did not reach statistical significance even after multivariate analysis (p = 0.977). The duration of PD was longer among osteoporotic patients, but this association did not reach statistical significance even after multivariate analysis (p = 0.921) (Table 3).

No statistically significant differences were observed between the two groups regarding biochemical parameters, including serum vitamin D, parathyroid hormone, albumin, and calcium. According to Spearman correlation, there was a significant positive correlation (p<0.01) between BMI and lumbar spine BMD and femoral neck BMD; higher patient BMI was associated with higher bone mineral density. A significant positive correlation was also identified between binder sevelamer supplementation and femoral neck BMD. Albumin and phosphate laboratory levels significantly correlated with BMD at the femoral neck (Table 4).

However, upon conducting multivariate analysis, only albumin demonstrated a significant correlation with BMD at the femoral neck (p<0.01) (Table 5).

## Discussion

In our study of 76 Peritoneal Dialysis patients, we observed osteoporosis prevalence of 41% and 38% based on femoral neck and lumbar spine T-scores, respectively. These figures diverge from findings in other studies; one study reported a lower prevalence of osteoporosis in PD patients, with 26% and 19% based on femoral neck and lumbar spine T-scores, respectively [22]. In contrast, another study found a higher prevalence of osteoporosis, with 78% at the femoral neck and 58% at the lumbar spine [30]. However, when considering the combined prevalence of osteoporosis and osteopenia in our study, the overall bone mineral loss in PD patients was as high as 52%, similar to what other studies have found [22]. Notably, we observed a significantly higher rate of osteoporosis and osteopenia in the femoral neck compared to the spine. This difference may be attributed to potential aortic calcifications, which

**Table 2. Univariate analysis of clinical and laboratory characteristics with BMD status.**

| | Osteoporosis (n = 76) | | |
|---|---|---|---|
| | *Yes (n = 40)* | *No (n = 36)* | P Value* |
| **Age** *Total* | 51 (18–79) | 52 (19–82) | 0.778 |
| **Gender** | | | 0.598 |
| Male | 23 (57.5%) | 18 (51.4%) | |
| Female | 17 (42.5%) | 17 (48.6%) | |
| **Body Mass Index** *Total* | 22.9 (16.9–29.3) | 26.9 (18.0–39.7) | 0.001* |
| **Duration of Peritoneal Dialysis** | 18.9 (5.3–67.2) | 17.7 (1.1–66.5) | 0.464 |
| **Hypertension** *(Yes)* | 31 (77.5%) | 26 (74.3%) | 0.745 |
| **Diabetes Mellitus** *(Yes)* | 13 (32.5%) | 12 (34.3%) | 0.870 |
| **Smoking History** *(Yes)* | 12 (30.0%) | 15 (42.9%) | 0.247 |
| **History of Hemodialysis** *(Yes)* | 34 (85.0%) | 26 (74.3%) | 0.247 |
| **History of Transplant** *(Yes)* | 7 (17.5%) | 5 (14.3%) | 0.705 |
| **History of Fractures** *(Yes)* | 5 (12.5%) | 1 (2.9%) | 0.125 |
| **Menopause Status** *(Yes)* | 10 (58.8%) | 8 (47.1%) | 0.492 |
| **Steroid Use** *(Yes)* | 3 (7.5%) | 1 (2.9%) | 0.372 |
| **Calcitriol Supplementation** *(Yes)* | 30 (75.0%) | 27 (77.1%) | 0.828 |
| **Calcium Supplementation** *(Yes)* | 23 (57.5%) | 20 (57.5%) | 0.975 |
| **Sevelamer Supplementation** *(Yes)* | 12 (30.0%) | 16 (45.7%) | 0.160 |
| **25 Vitamin D** *(ng/ml)* | 8.9 (3.0–34.9) | 8.9 (3.0–33.26) | 0.907 |
| **Alkaline Phosphatase** *(U/L)* | 106.0 (59.0–443.0) | 104.0 (48.0–204.0) | 0.162 |
| **Parathyroid Hormone** *(pg/ml)* | 297.7 (16.73–1525.0) | 388.5 (72.5–1372.0) | 0.260 |
| **Albumin** *(g/dl)* | 3.6 (2.5–4.3) | 3.7 (2.6–4.5) | 0.086 |
| **Calcium Measured** *(mg/dl)* | 9.1 (7.1–10.9) | 8.9 (7.3–10.4) | 0.758 |
| **Phosphate** *(mg/dl)* | 5.3 (1.6–7.8) | 5.8 (2.8–9.9) | 0.051 |
| **Ferritin** *(ng/ml)* | 468.5 (17.0–2547.0) | 309.0 (15.0–1988.0) | 0.640 |

*Mann–Whitney U test, Chi-Square test, aP Value: adjusted P value, aOR: adjusted Odds Ratio, CI: Confidence Interval

**Table 3. Multivariate analysis of clinical and laboratory characteristics with BMD status.**

| | aP Value | aOR (95% CI) |
|---|---|---|
| **Age** | 0.108 | 1.04 (0.99–1.09) |
| **Gender** *(Ref: male)* | 0.977 | 0.98 (0.25–3.87) |
| **Body Mass Index** | 0.001* | 0.77 (0.66–0.89) |
| **Duration of Peritoneal Dialysis** | 0.921 | 0.99 (0.96–1.04) |
| **Calcitriol Supplementation** *(Ref: No)* | 0.656 | 1.39 (0.33–5.88) |
| **Calcium Supplementation** *(Ref: No)* | 0.623 | 1.00 (0.99–1.00) |
| **Sevelamer Supplementation** *(Ref: No)* | 0.308 | 1.00 (0.99–1.00) |
| **25 Vitamin D** *(ng/ml)* | 0.619 | 1.02 (0.94–1.12) |
| **Alkaline Phosphatase** *(U/L)* | 0.045* | 1.02 (1.00–1.04) |
| **Parathyroid Hormone** *(pg/ml)* | 0.383 | 0.99 (0.99–1.00) |
| **Albumin** *(g/dl)* | 0.741 | 0.76 (0.15–3.96) |
| **Calcium Measured** *(mg/dl)* | 0.659 | 1.25 (0.47–3.33) |
| **Phosphate** *(mg/dl)* | 0.240 | 0.72 (0.41–1.25) |

***Ref**: Reference group; **aP** Value: adjusted P value; **aOR**: adjusted Odds Ratio; **CI**: Confidence Interval

**Table 4. Spearman correlation of clinical and laboratory variables with BMD status.**

| | BMD Femoral | | BMD Lumbar | |
|---|---|---|---|---|
| | Correlation Coefficient | P-value | Correlation Coefficient | P-value |
| Age | -0.56 | 0.635 | 0.117 | 0.319 |
| Body Mass Index | 0.479 | 0.01* | 0.465 | 0.01* |
| Height | -0.041 | 0.730 | 0.113 | 0.335 |
| Weight | 0.450 | 0.01* | 0.525 | 0.01* |
| Duration of PD | -0.116 | 0.321 | -0.041 | 0.730 |
| Calcitriol Dose | 0.188 | 0.106 | -0.016 | 0.894 |
| Calcium Dose | 0.042 | 0.718 | 0.036 | 0.760 |
| Sevelamer Dose | 0.246 | 0.033* | 0.124 | 0.288 |
| Vitamin D level | 0.047 | 0.689 | 0.048 | 0.684 |
| Alkaline Phosphatase | -0.088 | 0.450 | -0.049 | 0.679 |
| Parathyroid Hormone | 0.216 | 0.063 | 0.149 | 0.201 |
| Albumin | 0.324 | 0.01* | 0.126 | 0.280 |
| Calcium | -0.004 | 0.972 | -0.101 | 0.390 |
| Phosphate | 0.357 | 0.01* | 0.180 | 0.123 |
| Ferritin | -0.101 | 0.389 | -0.181 | 0.121 |

**PD**: Peritoneal Dialysis

could lead to an overestimation of the spine BMD. However, one recent study showed that aortic vascular calcification had a minimal impact on the lumbar spine BMD [31]. Another possible explanation might be that cortical bone appears to be affected more than trabecular bone in renal osteodystrophy. Given the significant presence of cortical bone in the femoral neck and hip, these areas may be more susceptible to bone density alterations.

**Table 5. Multiple linear regression analysis of laboratory variables with BMD status.**

| | BMD Femoral | | BMD Lumbar | |
|---|---|---|---|---|
| | Beta (95%CI) | aP-value | Beta (95%CI) | aP-value |
| Age | | | | |
| Body Mass Index | 0.428 (0.05–0.13) | 0.001* | 0.445 (0.06–0.17) | 0.001* |
| Height | | | | |
| Weight | | | | |
| Duration of PD | | | | |
| Calcitriol Dose | | | | |
| Calcium Dose | | | | |
| Sevelamer Dose | 0.171 (0.00–0.00) | 0.096 | 0.062 (0.00–0.00) | 0.585 |
| Vitamin D level | | | | |
| Alkaline Phosphatase | | | | |
| Parathyroid Hormone | | | | |
| Albumin | 0,279 (0.21–1.30) | 0.007* | 0.105 (-0.41–1.12) | 0.356 |
| Calcium | | | | |
| Phosphate | 0.166 (-0.04–0.28) | 0.138 | 0.126 (-0.11–0.34) | 0.314 |
| Ferritin | | | | |

**aP**-value: adjusted P-value; **CI**: Confidence interval; **PD**: Peritoneal Dialysis

Moreover, we identified additional correlations, including albumin, phosphorus, and sevelamer use, specifically with femoral BMD instead of lumber BMD. This observation may indicate a stronger correlation with cortical bone-rich areas [32]. Additionally, cortical bone has also been linked to mortality, as evidenced by a five-year follow-up study in HD patients, which demonstrated that low BMD at the hip predicted mortality [33, 34].

In our study, 86.8% of our patients were found to have 25 vitamin D deficiency, with a median of 8.7ng/mL. One multicenter study conducted on PD patients reported similar results [15]. However, studies conducted on HD patients found higher levels of 25 vitamin D compared to our findings [35]. This may be explained by the peritoneal losses of the active vitamin D sterols that occur in peritoneal dialysis [15]. Interestingly, we did not observe any significant correlation between 25 vitamin D levels and BMD in the multivariate analysis. However, a recent study evaluating 50 HD patients showed no significant association between vitamin D levels and BMD status [36].

In the general population, the incidence of osteoporosis is typically expected to increase with older age and female sex. Surprisingly, in our study, the prevalence of osteoporosis was higher in male subjects (57.5%) compared to females (42.5%). However, this difference was not statistically significant. We could not establish a significant correlation between age and BMD in our patient population. BMI, however, demonstrated a robust positive correlation with BMD values as measured by DEXA. Similar to a previous study, we observed that patients' weight, rather than height, was responsible for the correlation between BMI and BMD [22]. This finding might be attributed to a certain level of measurement error with DEXA scans in individuals with obesity [37, 38]. Hence, the reliability of BMI-BMD correlations in the lumbar spine region may be questioned.

In contrast, the femoral neck is predicted to have less fat tissue due to its anatomical location. Our study observed a similarly strong correlation between BMI and BMD measurements in the femoral neck area. Therefore, we also predict that the positive correlation between BMI and BMD is not just a measurement error but a significant clinical correlation. Obesity has been shown to be associated with higher BMD of weight-bearing bones, suggesting that obesity may contribute to preventing bone loss in such bones, pointing to the significance of reduced body weight as the primary risk factor for decreased BMD in individuals undergoing chronic PD.

Previous research has shown that low BMD is linked to prolonged dialysis periods, elevated serum PTH levels, higher serum calcium levels, and a decrease in the consumption of elemental calcium supplements [22, 30]. In a prior study involving 194 dialysis patients, we demonstrated a statistically significant negative correlation between BMD and duration of dialysis. Furthermore, multivariate analysis identified an association between PTH levels and femoral BMD as assessed by DEXA scan [21]. However, in this study, we could not replicate similar results in our patients' population. This might be attributed to our failure to account for the entire duration of dialysis, which includes both PD and HD time. Such inclusion could reveal a significant correlation. We did observe a positive correlation between PTH and BMD at the femoral neck using Spearman's correlation, with a p-value of 0.063. However, it is worth noting that our relatively small sample size may also influence this finding; with a larger sample size, a statistically significant correlation with PTH might have been observed, especially considering that a subset of patients experiences secondary/tertiary hyperparathyroidism, which could potentially explain the low BMD in these individuals.

Nevertheless, osteoporotic individuals exhibited higher alkaline phosphatase levels, a marker indicating bone turnover, consistent with findings from previous research [39, 40]. However, although this observation reached statistical significance, it is essential to clarify that the slight difference in median alkaline phosphatase levels between osteoporotic and non-

osteoporotic groups may not be clinically significant. Spearman's correlation showed a positive correlation between femoral BMD and albumin and phosphorous levels; this was also evident in our prior study involving mainly HD patients. However, after the multivariate analysis, only albumin showed a statistically significant correlation with BMD at the femoral neck. This finding might be related to the nutritional status of the patients, as lower albumin levels may indicate poor nutritional status [41].

There are several potential limitations to this study. Firstly, using a cross-sectional study design indicates a definitive cause-and-effect relationship between variables of interest cannot be established. Secondly, the study was conducted at a solitary clinical center, so we must exercise caution when generalizing the findings. Thirdly, the sample size was limited to only 76 patients, which may need to be increased to draw conclusive relationships between variables of interest. Fourthly, relying solely on DEXA scan results for diagnosing osteoporosis may not be universally applicable, particularly in younger patients. Lastly, we did not account for the entire duration of dialysis, which includes both PD and HD time.

Further studies are needed with a larger sample size and a longer time to study the variables in more detail. However, despite these limitations, this study is the first in the region to explore Bone Mineral Density and its relationship with vitamin D levels and other biochemical factors in peritoneal dialysis patients. This is particularly noteworthy considering the scarcity of research on this subject. Furthermore, our study, with its meticulous data collection, rigorous statistical analyses, comprehensive exploration of various variables, and thoughtful consideration of potential confounders, enhances the reliability and significance of our findings. This contributes to advancing our understanding of Bone Mineral Density and related variables in patients undergoing peritoneal dialysis.

## Conclusion

In conclusion, this study provides valuable insights into the prevalence of osteoporosis and osteopenia among the region's Peritoneal Dialysis (PD) patients. A significant proportion of PD patients exhibited reduced Bone Mineral Density (BMD), with osteoporosis rates of 41% in the femoral neck and 38% in the lumbar spine. While our study did not establish a significant correlation between vitamin D levels and BMD in PD patients, it highlighted this group's high prevalence of vitamin D deficiency. Furthermore, our analysis revealed a strong positive correlation between BMI and BMD, particularly in the femoral neck area.

Addressing bone health in PD patients is paramount to mitigating the risk of fractures and improving the overall quality of life. Healthcare providers should consider integrating regular BMD assessments and vitamin D supplementation into the comprehensive care plan for CKD patients undergoing PD. This study serves as a foundation for future investigations in this field, with the ultimate goal of improving the management and outcomes of bone health in PD patients.

## Supporting information

**S1 Data.**
(XLSX)

## Acknowledgments

We acknowledge all of the patients who participated in the study. We acknowledge the staff at Al-Rahma Medical Center for performing DEXA scans required for our patients. We also

acknowledge staff in the An-Najah National University Hospital dialysis center for managing patients and keeping follow-ups.

## Author Contributions

**Conceptualization:** Mazen Abdalla, Zakaria Hamdan, Zaher Nazzal.

**Data curation:** Rami Tamimi, Amjad Bdair, Ahmad Shratih, Alaa Sarsour.

**Formal analysis:** Rami Tamimi, Amjad Bdair, Ahmad Shratih, Zaher Nazzal.

**Investigation:** Rami Tamimi.

**Methodology:** Amjad Bdair, Ahmad Shratih, Mazen Abdalla, Alaa Sarsour, Zakaria Hamdan, Zaher Nazzal.

**Project administration:** Zaher Nazzal.

**Resources:** Zaher Nazzal.

**Supervision:** Mazen Abdalla, Zakaria Hamdan, Zaher Nazzal.

**Validation:** Alaa Sarsour, Zakaria Hamdan.

**Writing – original draft:** Rami Tamimi, Amjad Bdair.

**Writing – review & editing:** Ahmad Shratih, Mazen Abdalla, Alaa Sarsour, Zakaria Hamdan, Zaher Nazzal.

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
