## [Decision Letter · Decision Letter 0]

8 Jan 2024

PONE-D-23-30185Bone Mineral Density and Related Clinical and Laboratory Factors in Peritoneal Dialysis Patients: Implications for Bone Health ManagementPLOS ONE

Dear Dr. Nazzal,

Thank you for submitting your manuscript to PLOS ONE. After careful consideration, we feel that it has merit but does not fully meet PLOS ONE’s publication criteria as it currently stands. Therefore, we invite you to submit a revised version of the manuscript that addresses the points raised during the review process.

We look forward to receiving your revised manuscript.

Kind regards,

Ali B. Roomi, Asst. Prof. Dr

Academic Editor

PLOS ONE

Journal Requirements:

Additional Editor Comments:

General Comments:

This is a single-center observational study that aims to determine the prevalence of osteoporosis in Peritoneal Dialysis patients using bone mineral density measurements obtained through dual-energy X-ray absorptiometry scans. The study also explores possible associations with clinical and biochemical factors.

Specific Revision Comments:

1. In the Introduction, please discuss the association between vitamin D and BMD found in kidney transplant recipients (KTRs), emphasizing the importance of early identification and correction of low BDM in peritoneal treatment patients who are often candidates for kidney transplants, adding the reference (Nutrients 2022, 14, 323. https://doi.org/10.3390/nu14020323).

2. In the Methods section, please consider addressing patients undergoing calcimimetic treatment. If included, add calcimimetics to the current medication history and consider them as a confounding variable in statistical analysis.

3. In the Study Measures section, please provide more details regarding the dosage of the parathormone kit.

4. In the Analysis Plan, please explicitly mention the variables used in regression analysis.

5. In the Results section, address the very low level of 25-hydroxyvitamin D. Explain how the authors interpret this data.

6. In Table 1, please define the history of the type of transplant being referred to.

7. In Table 1, please include data regarding inactive vitamin D supplementation.

8. In Figure 1, please ensure that the groups are consistently divided into non-osteoporosis and osteoporosis, as the figure currently reports data categorized into three groups. Please, homogenize the data.

9. In the Background Characteristics of the Patients (line 157, page 8), pleae note that the figure 1 in the brackets (Bone Mineral Density Distribution) does not match with the text (vitamin D lab results).

10. Tables 2 and 3: Divide Univariate and Multivariate analyses into two separate tables.

11. In the Discussion, please highlight the strengths of this study.

12. Please perform an extensive English language revision.

Reviewers' comments:

Reviewer's Responses to Questions

**Comments to the Author**

1. Is the manuscript technically sound, and do the data support the conclusions?

Reviewer #1: Yes

Reviewer #2: No

Reviewer #3: Partly

2. Has the statistical analysis been performed appropriately and rigorously? 

Reviewer #1: Yes

Reviewer #2: Yes

Reviewer #3: No

3. Have the authors made all data underlying the findings in their manuscript fully available?

Reviewer #1: No

Reviewer #2: Yes

Reviewer #3: Yes

4. Is the manuscript presented in an intelligible fashion and written in standard English?

Reviewer #1: Yes

Reviewer #2: Yes

Reviewer #3: No

5. Review Comments to the Author

Reviewer #1: Dear editor, thank you for alllwing me the opportunity to review this paper. Overal, it is nicely crafted and reads well. The findings are not novel although data on PD patients a relatively scanty. The reviewer has not special comments to make.

Reviewer #2: This study evaluates end-stage renal disease (ESRD) patients and risk of bone loss. The authors claim that the patients have accelerated bone turnover, leading to osteoporosis and osteopenia.

The authors enrolled 76 peritoneal dialysis patients and performed DXA scan to evaluate BMD 31 at classic sites. By multivariate analysis they describe the relationship between BMD and clinical and biochemical parameters. The authors observed about 50% of the patients had low bone mass i.e. BMD T-score below -2,5 and formulated as osteoporosis and not surprising with a higher prevalence among patients with low BMI. Due to higher bone turnover higher alkaline phosphatase levels were observed among patients with low bone mass and vitamin D deficiency was also prevalent affecting about 85% of patients.

The authors conclude correctly that they have observed that a notable proportion of PD patients experience reduced BMD and high degree of low vitamin D and the low BMD was correlated to low BMI.

I do not find these observations new or surprising. It is very well known that high bone turnover increased alkaline phosphatase. That low BMI increase the risk of low bone mass and that vitamin D deficiency leads to immature bone and low BMD.

I dislike the use of the term osteoporosis in the group of patients described and prefer the use of T-scores/Z-scores when younger than 50 years of age. The patients are very heterogenous i.e. some have had fractures, some are in prednisolone, they are male and females and aged younger and older than 50 years of age.

A significant amount of the patients are suffering of secondary/tertiary hyperparathyroidism and the authors do not describe or discuss the impact hereof. The low bone mass might be explained by the renal disease (osteomalcia, hyperparathyroidism and renal ostedystrophia). Due to that the use of the term osteoporosis are misleading although all of the above might lead to increased risk of fractures.

The reported alkaline phosphatase values are just slightly increase (mean value) although correlates to low bone mass. What does than mean in a clinical setting of the use of biochemistry to pin-point the diagnosis of low bone mass – claimed by the authors themselves?

The authors cannot conclude anything from the present study other than it seems that PD patients are also of risk of low bone mass for many reasons and thereby might have an increased risk of fractures. That sufficient care of keeping normal vitamin D among other recommendations are needed. However, it is unknown if this would decrease the risk of low BMD.

Reviewer #3: General Comments:

This is a single-center observational study that aims to determine the prevalence of osteoporosis in Peritoneal Dialysis patients using bone mineral density measurements obtained through dual-energy X-ray absorptiometry scans. The study also explores possible associations with clinical and biochemical factors.

Specific Revision Comments:

1. In the Introduction, please discuss the association between vitamin D and BMD found in kidney transplant recipients (KTRs), emphasizing the importance of early identification and correction of low BDM in peritoneal treatment patients who are often candidates for kidney transplants, adding the reference (Nutrients 2022, 14, 323. https://doi.org/10.3390/nu14020323).

2. In the Methods section, please consider addressing patients undergoing calcimimetic treatment. If included, add calcimimetics to the current medication history and consider them as a confounding variable in statistical analysis.

3. In the Study Measures section, please provide more details regarding the dosage of the parathormone kit.

4. In the Analysis Plan, please explicitly mention the variables used in regression analysis.

5. In the Results section, address the very low level of 25-hydroxyvitamin D. Explain how the authors interpret this data.

6. In Table 1, please define the history of the type of transplant being referred to.

7. In Table 1, please include data regarding inactive vitamin D supplementation.

8. In Figure 1, please ensure that the groups are consistently divided into non-osteoporosis and osteoporosis, as the figure currently reports data categorized into three groups. Please, homogenize the data.

9. In the Background Characteristics of the Patients (line 157, page 8), pleae note that the figure 1 in the brackets (Bone Mineral Density Distribution) does not match with the text (vitamin D lab results).

10. Tables 2 and 3: Divide Univariate and Multivariate analyses into two separate tables.

11. In the Discussion, please highlight the strengths of this study.

12. Please perform an extensive English language revision.

6. PLOS authors have the option to publish the peer review history of their article (what does this mean?). If published, this will include your full peer review and any attached files.

Reviewer #1: No

Reviewer #2: No

Reviewer #3: No

---

## [Author Response · Author response to Decision Letter 0]

2 Feb 2024

Reviewer 1 comments:

Dear editor, thank you for allowing me the opportunity to review this paper. Overall, it is nicely crafted and reads well. The findings are not novel although data on PD patients a relatively scanty. The reviewer has not special comments to make.

Authors’ Response: Thank you for your time reading and evaluating the manuscript. We agree with you that some findings within our study may not be deemed novel. However, our primary objective was to evaluate the precise extent of BMD reduction within this specific patient demographic, thereby laying a foundation for subsequent investigations. Moreover, the existing literature lacks consistency in detailing the correlations between clinical and biochemical factors and BMD, underscoring the significance of our efforts to contribute clarity and insight to this nuanced discussion.

Reviewer 2 comments

This study evaluates end-stage renal disease (ESRD) patients and risk of bone loss. The authors claim that the patients have accelerated bone turnover, leading to osteoporosis and osteopenia.

The authors enrolled 76 peritoneal dialysis patients and performed DXA scan to evaluate BMD 31 at classic sites. By multivariate analysis they describe the relationship between BMD and clinical and biochemical parameters. The authors observed about 50% of the patients had low bone mass i.e. BMD T-score below -2,5 and formulated as osteoporosis and not surprising with a higher prevalence among patients with low BMI. Due to higher bone turnover higher alkaline phosphatase levels were observed among patients with low bone mass and vitamin D deficiency was also prevalent affecting about 85% of patients.

The authors conclude correctly that they have observed that a notable proportion of PD patients experience reduced BMD and high degree of low vitamin D and the low BMD was correlated to low BMI.

Authors’ Response: Thank you for your time reading and evaluating the manuscript. We appreciate your supportive comments, they were very helpful and we believe they enhanced the manuscript. We carefully addressed each comment individually and incorporated relevant adjustments accordingly.

I do not find these observations new or surprising. It is very well known that high bone turnover increased alkaline phosphatase. That low BMI increase the risk of low bone mass and that vitamin D deficiency leads to immature bone and low BMD.

Authors’ Response: We expected certain findings for some factors. For instance, we expected BMD to exhibit a low measure among PD patients. However, our primary objective was to evaluate the precise extent of BMD reduction as well as Vitamin D deficiency within this specific patient demographic, thereby laying a foundation for subsequent investigations. Moreover, the existing literature lacks consistency in detailing the correlations between clinical and biochemical factors and BMD. For example, we expected certain results and associations for some biochemical factors, but the data didn't match those expectations.

I dislike the use of the term osteoporosis in the group of patients described and prefer the use of T-scores/Z-scores when younger than 50 years of age. The patients are very heterogenous i.e. some have had fractures, some are in prednisolone, they are male and females and aged younger and older than 50 years of age.

Authors’ Response: We agree to your point of view. For simplicity and convenience, we used the term "osteoporosis" when T-scores fell below -2.5. However, we acknowledge in the study limitations that relying solely on DEXA scan results for diagnosing osteoporosis may not be applicable to all patients' categories.

A significant amount of the patients are suffering of secondary/tertiary hyperparathyroidism and the authors do not describe or discuss the impact hereof. The low bone mass might be explained by the renal disease (osteomalcia, hyperparathyroidism and renal ostedystrophia). Due to that the use of the term osteoporosis are misleading although all of the above might lead to increased risk of fractures.

Authors’ Response: Since we employed DEXA scans for bone assessment, we chose to categorize patients into osteoporosis and osteopenia based on T scores. While acknowledging renal osteodystrophy's precision, our choice aligns with the DEXA scan methodology for bone status evaluation. Despite a notable portion of patients experiencing secondary/tertiary hyperparathyroidism, our multivariate analysis, encompassing both parathyroid hormone (PTH) and calcium laboratory results, failed to discern any significant association between these factors and bone mineral density (BMD), as expressed by T-score measurements. We elaborated on this point in the discussion section.

The reported alkaline phosphatase values are just slightly increase (mean value) although correlates to low bone mass. What does that mean in a clinical setting of the use of biochemistry to pin-point the diagnosis of low bone mass – claimed by the authors themselves?

Authors’ Response: In this study we don’t claim that elevated alkaline phosphatase levels can be utilized as a means to diagnose low bone mass in PD patients. Rather, we claim that higher alkaline phosphatase levels were observed in patients with lower BMD, and this observation reached statistical significance. It's important to clarify that while the statistical significance is established, we do not claim it to be clinically significant. We elaborated on this point more detail in the discussion section.

The authors cannot conclude anything from the present study other than it seems that PD patients are also of risk of low bone mass for many reasons and thereby might have an increased risk of fractures. That sufficient care of keeping normal vitamin D among other recommendations are needed. However, it is unknown if this would decrease the risk of low BMD.

Authors’ Response: In this cross-sectional study, we determined the prevalence rates of osteoporosis and osteopenia among PD patients in the region, along with the prevalence of vitamin D deficiency. While PD patients are inherently at risk of low bone mass for various reasons, our findings revealed that certain factors expected to influence bone health, such as PTH levels, Ca levels, Vitamin D levels, age, and gender, did not exhibit a significant role, at least in this study. It is noteworthy that future studies exploring the change of these variables within this population may alter the prevalence of bone mineral density. For example, a future trial of Vitamin D treatment with a pre and post DEXA scan to evaluate the effect of vitamin D in BMD changes among PD patients.

Reviewer 3 comments

1. In the Introduction, please discuss the association between vitamin D and BMD found in kidney transplant recipients (KTRs), emphasizing the importance of early identification and correction of low BDM in peritoneal treatment patients who are often candidates for kidney transplants, adding the reference (Nutrients 2022, 14, 323. https://doi.org/10.3390/nu14020323).

Authors’ Response: Thank you for your suggestion. Case discussed in the introduction of the manuscript (line 66-68).

2. In the Methods section, please consider addressing patients undergoing calcimimetic treatment. If included, add calcimimetics to the current medication history and consider them as a confounding variable in statistical analysis

Authors’ Response: Patients in this cross-sectional study were not on any calcimimetic treatment.

3. In the Study Measures section, please provide more details regarding the dosage of the parathormone kit

Authors’ Response: The kit used to determine the levels of Parathyroid hormone was added 

4. In the Analysis Plan, please explicitly mention the variables used in regression analysis

Authors’ Response: Variables used in the regression analysis were added.

5. In the Results section, address the very low level of 25-hydroxyvitamin D. Explain how the authors interpret this data

Authors’ Response: The findings on 25-hydroxyvitamin D are addressed in the results section and interpreted in the discussion section.

6. In Table 1, please define the history of the type of transplant being referred to.

Authors’ Response: History of kidney transplant added.

7. In Table 1, please include data regarding inactive vitamin D supplementation.

Authors’ Response: Patients in this study were only on active Vitamin D supplementation (alpha D3). None were on inactive Vit D (cholecalciferol).

8. In Figure 1, please ensure that the groups are consistently divided into non-osteoporosis and osteoporosis, as the figure currently reports data categorized into three groups. Please, homogenize the data.

Authors’ Response: Thank you for noting this. The figure has been corrected.

9. In the Background Characteristics of the Patients (line 157, page 8), please note that the figure 1 in the brackets (Bone Mineral Density Distribution) does not match with the text (vitamin D lab results).

Authors’ Response: This typo was unintentional, and it has been rectified. The figure 1 has been placed in its correct position.

10. Tables 2 and 3: Divide Univariate and Multivariate analyses into two separate tables

Authors’ Response: We agree with your suggestion. We divided Univariate and Multivariate analyses into two separate tables; Tables 2, 3, 4, and 5.

11. In the Discussion, please highlight the strengths of this study.

Authors’ Response: Strengths of the study were highlighted.

12. Please perform an extensive English language revision.

Authors’ Response: Extensive English language revision was done.

---

## [Decision Letter · Decision Letter 1]

25 Mar 2024

Bone Mineral Density and Related Clinical and Laboratory Factors in Peritoneal Dialysis Patients: Implications for Bone Health Management

PONE-D-23-30185R1

Dear Dr. Nazzal,

We’re pleased to inform you that your manuscript has been judged scientifically suitable for publication and will be formally accepted for publication once it meets all outstanding technical requirements.

Kind regards,

Patricia Khashayar

Academic Editor

PLOS ONE

Additional Editor Comments (optional):

Reviewers' comments:

Reviewer's Responses to Questions

**Comments to the Author**

1. If the authors have adequately addressed your comments raised in a previous round of review and you feel that this manuscript is now acceptable for publication, you may indicate that here to bypass the “Comments to the Author” section, enter your conflict of interest statement in the “Confidential to Editor” section, and submit your "Accept" recommendation.

Reviewer #1: All comments have been addressed

Reviewer #3: All comments have been addressed

2. Is the manuscript technically sound, and do the data support the conclusions?

Reviewer #1: Yes

Reviewer #3: Yes

3. Has the statistical analysis been performed appropriately and rigorously? 

Reviewer #1: Yes

Reviewer #3: Yes

4. Have the authors made all data underlying the findings in their manuscript fully available?

Reviewer #1: Yes

Reviewer #3: Yes

5. Is the manuscript presented in an intelligible fashion and written in standard English?

Reviewer #1: Yes

Reviewer #3: Yes

6. Review Comments to the Author

Reviewer #1: No further comments. Authors have addressed all comments in an adequate manner.

Reviewer #3: The authors have satisfactorily addressed all the concerns I had with the previous submission and made the necessary changes in the manuscript.

7. PLOS authors have the option to publish the peer review history of their article (what does this mean?). If published, this will include your full peer review and any attached files.

Reviewer #1: No

Reviewer #3: No

---

## [Editor Report · Acceptance letter]

3 May 2024

PONE-D-23-30185R1 

PLOS ONE

Dear Dr. Nazzal, 

I'm pleased to inform you that your manuscript has been deemed suitable for publication in PLOS ONE. Congratulations! Your manuscript is now being handed over to our production team.

Kind regards, 

on behalf of

Dr. Patricia Khashayar 

Academic Editor

PLOS ONE